# Prediction interval for neural network models using weighted asymmetric loss functions.

## Abstract

We propose a simple and efficient approach to generate prediction intervals (PIs) for approximated and forecasted trends. Our method leverages a weighted asymmetric loss function to estimate the lower and upper bounds of the PIs, with the weights determined by the interval width. We provide a concise mathematical proof of the method, show how it can be extended to derive PIs for parametrised functions and argue why the method works for predicting PIs of dependent variables. The presented tests of the method on a real-world forecasting task using a neural network-based model show that it can produce reliable PIs in complex machine learning scenarios.

## 1 Introduction

Neural network models are increasingly often used in prediction tasks, for example in weather Liu et al. (2021), water level Zhang et al. (2015), price Duan et al. (2022), electricity grid load Rana et al. (2013), ecology Miok (2018), demographics Werpachowska (2018) or sales forecasting. However, their often cited weakness is that—in their vanilla form—they provide only point predictions. Meanwhile, many of their users are interested also in prediction intervals (PIs), that is, ranges $[l, u]$ containing forecasted values with a given probability (e.g. 95%).

Several approaches have been proposed to facilitate the estimation of PIs (see Beven & Binley (1992); MacKay (1992); Nix & Weigend (1994); Chryssolouris et al. (1996); Heskes (1996); Hwang & Ding (1997); Zapranis & Livanis (2005); Khosravi et al. (2010); Rana et al. (2013); Zhang et al. (2015); Miok (2018); Liu et al. (2021) and references therein):

1. the delta method, which assumes that prediction errors are homogeneous and normally distributed Chryssolouris et al. (1996); Hwang & Ding (1997);

2. Bayesian inference MacKay (1992), which requires a detailed model of sources of uncertainty, and is extremely expensive computationally for realistic forecasting scenarios Khosravi et al. (2010);

3. Generalized Likelihood Uncertainty Estimation (GLUE) Beven & Binley (1992), which requires multiple runs of the model with parameters sampled from a distribution specified by the modeller;

4. bootstrap Heskes (1996), which generates multiple training datasets, leading to high computational cost for large datasets Khosravi et al. (2010);

5. Mean-Variance Estimation (MVE) Nix & Weigend (1994), which is less computationally demanding than the methods mentioned above but also assumes a normal distribution of errors and gives poor results Khosravi et al. (2010);

6. Lower Upper Bound Estimation (LUBE), which trains the neural network model to directly generate estimations of the lower and upper bounds of the prediction interval using a specially designed training procedure with tunable parameters Khosravi et al. (2010); Rana et al. (2013); Zhang et al. (2015); Liu et al. (2021).

The existing methods are either overly restrictive (the delta method, MVE) or too computationally expensive. We propose a method which is closest in spirit to LUBE (we train a model to predict either a lower or an upper bound for the PI) but simpler and less computationally expensive, because it does not require any parameter tuning.

## 2 Problem statement

We consider a prediction problem $x \mapsto y$, where $x \in \mathcal{X}$ are features (e.g. $x \in \mathbb{R}^d$) and $y \in \mathbb{R}$ is the predicted variable. We assume that observed data $\mathcal{D} := \{(x, y)\}^N \subset \mathcal{X} \times \mathbb{R}$ are statistically independent $N$-realisations of a pair of random variables $(X, Y)$ with an unknown joint distribution $\mathcal{P}$. We also consider a model $g_\theta$ which, given $x \in \mathcal{X}$, produces a prediction $g_\theta(x)$, where $\theta$ is are model parameters in parameter space $\Theta = \mathbb{R}^m$. When forecasting, the prediction is also a function of an independent "time" variable $t$, which is simply included in $\mathcal{X}$.

The standard model training procedure aims to find such $\theta$ that, given $x \in \mathcal{X}$, $g_\theta(\cdot)$ is an good point estimate of $Y|X$, e.g. $g_\theta(x) \approx \mathbb{E}[Y|X = x]$. This is achieved by minimising a loss function—by abuse of notation—of the form $l(y, y') = l(y - y')$ with a minimum at $y = y'$ and increasing sufficiently fast for $|y - y'| \to \infty$, where $y$ is the observed target value and $y'$ is the model prediction. More precisely, we minimise the sample average of the loss function $l$ over the parameters $\theta$:

$$\hat{\theta} = \arg\min_\theta \sum_{i=1}^N l\left(y_i, g_\theta(x_i)\right) \ .$$

The above procedure can be given a simple probabilistic interpretation by assuming that the target value $y$ is a realisation of a random variable $Y$ with the distribution $\mu + Z$, where $\mu$ is an unknown "true value" and $Z$ is an i.i.d. error term with a probability density function $\rho(z) \sim \exp(-l(z))$. Two well-known functions, Mean Squared Error (MSE) and Mean Absolute Error (MAE), correspond to assuming a Gaussian or Laplace distribution for $Z$, respectively. The value which minimises the loss function $l$ corresponds then to the maximum log-likelihood estimation of the unknown parameter $\mu$, since $\ln P(y|\mu) \sim -l(y - \mu)$.

In this paper we focus on the MAE, in which case the average loss function $l$ (i.e. negative log-likelihood of data $\mathcal{D}$)

$$l(y, y') = |y - y'|$$

Given an i.i.d. sample $\{y_i\}_{i=1}^N$, we thus try to minimise

$$\frac{1}{N} \sum_{i=1}^N |y_i - y'|$$

which for $N \to \infty$ equals $\mathbb{E}[|Y - y'|]$. The *optimal* value of $y'$, i.e. the value which minimises the loss, noted as $\hat{y}$, equals

$$\hat{y} = \arg\min_{y' \in \mathbb{R}} \mathbb{E}[|Y - y'|] \ \text{ for } Y = \mu + L \ ,$$

where $L$ has Laplace distribution with density $\rho_L(z) = e^{-|z|}/2$. The minimum fulfills the condition $\partial \mathbb{E}[|Y - y'|]/\partial y' = 0$. Since

$$\mathbb{E}[|Y - y'|] = \mathbb{E}[|\mu + L - y'|]$$

$$= \frac{1}{2} \int_{y'-\mu}^\infty e^{-|z|}(z + \mu - y')dz - \frac{1}{2} \int_{-\infty}^{y'-\mu} e^{-|z|}(z + \mu - y')dz \ ,$$

we have

$$\frac{\partial \mathbb{E}[|Y - y'|]}{\partial y'} = \frac{1}{2} \int_{-\infty}^{y'-\mu} e^{-|z|}dz - \frac{1}{2} \int_{y'-\mu}^\infty e^{-|z|}dz \ ,$$

which is zero iff $y' - \mu = 0$, hence $\hat{y} = \mu$. For a *finite* sample $[y_i]_{i=1}^N$, $\hat{y}$ is the sample median, which approaches $\mu$ as $N \to \infty$.

In prediction, we work with an independent variable $X$ and a dependent variable $Y$, under the assumption that there exists some mapping $g$ such that $g(X) = Y + \epsilon$ with some error $\epsilon$. We aim to find a prediction interval such that given an $x$, the predicted value $y$ lies within this interval with probability $p$. Note that this problem is equivalent to finding the $\alpha_l$-th and $\alpha_u$-th percentile of the distribution $Y|X$ , such that $0 \leq \alpha_l \leq \alpha_u \leq 1$ and $\alpha_u - \alpha_l = p$. These percentiles then correspond to the lower or upper bound of the PI. To this end, we are going to generalise the above result and train the model to predict a desired percentile of the distribution $Y|X$.

## 3   Main result

We show semi-analytically that the prediction interval can be calculated in an efficient way by training the model minimising a weighted, i.e. not symmetric around 0, MAE loss function:

$$l(y - y') = 2 \left[ \alpha(y - y')^+ + (1 - \alpha)(y - y')^- \right] ,$$

for $\alpha \in (0, 1)$. The factor of 2 is introduced to rescale the defined loss to match the standard MAE definition.

The main mathematical result is derived from the insight presented in our Theorem 1 below. It relates the choice of loss function to the resulting corresponding minimizer, for a specific class of loss functions.

**Theorem 1.** *For some $\alpha \in (0, 1)$, let $l_\alpha : \mathbb{R}^2 \to \mathbb{R}$ be a loss function, defined by*

$$l_\alpha(y, \hat{y}) = \begin{cases} 2(1 - \alpha)|y - \hat{y}| & \text{for } y - \hat{y} < 0 \\ 2\alpha|y - \hat{y}| & \text{otherwise} \end{cases} \tag{1}$$

*Let $Y$ be a random variable on probability space $(\Omega, \mathcal{F}, \mathbb{P})$ with distribution $\mu_Y := Y\#\mathbb{P}$. Then, the value $\hat{y}$ which minimises $\mathbb{E}\left[l_\alpha(Y, \hat{y})\right]$ is the $\alpha$-th percentile of $Y$.*

*Proof.* The goal is to find $\hat{y}$, which minimises $\mathbb{E}\left[l_\alpha(Y, \hat{y})\right]$. For this $\hat{y}$, we have

$$
\begin{aligned}
0 &= \frac{\partial}{\partial \hat{y}} \mathbb{E}[l_\alpha(Y, \hat{y})] \\
&= \frac{\partial}{\partial \hat{y}} \left[ \int_{-\infty}^{\hat{y}} -(1 - \alpha)(y - \hat{y}) \, \mathrm{d}\mu_Y(y) + \int_{\hat{y}}^{\infty} \alpha(y - \hat{y}) \, \mathrm{d}\mu_Y(y) \right] \\
&= (1 - \alpha) \int_{-\infty}^{\hat{y}} \mathrm{d}\mu_Y(y) - \alpha \int_{\hat{y}}^{\infty} \mathrm{d}\mu_Y(y) \\
&= (1 - \alpha)F_Y(\hat{y}) - \alpha(1 - F_Y(\hat{y})) ,
\end{aligned}
\tag{2}
$$

where $F_Y(\hat{y}) = \int_{-\infty}^{\hat{y}} \mathrm{d}\mu_Y(y)$ is the CDF of $Y$ and we neglect the constant factor of 2. Hence, $\hat{y} = F_Y^{-1}(\alpha)$. □

**Remark 1.** *While the derivation works for any distribution $\mu$ supported on the real axis, the interpretation of MAE minimisation as maximum likelihood estimation of $\hat{y}$ for $\alpha = 1/2$ is valid only when $Y$ has a Laplace distribution. Note that our result does not require assuming an independent Laplace distribution for the error term.*

**Remark 2.** *An analogous calculation shows that minimising $\sum_i l(y_i, \hat{y})$ over $\hat{y}$ leads to $\hat{y}$ equal to the $\alpha$-th percentile of the sample $\{y_i\}$.*

**Corollary 2.** *If $\hat{y}$ is restricted to $\hat{y} = g(\theta)$ for some differentiable function $g : \mathbb{R}^d \to \mathbb{R}$, the theorem still holds assuming that $\partial g(\theta)/\partial \theta \neq 0$ for all $\theta$.*

**Remark 3.** *Mind that $\partial g(\theta)/\partial \theta$ is a vector, as it is a gradient rather than a scalar derivative, and thus only one of its components must be non-zero.*

*Proof.* In the proof of Theorem 1, we replace $y$ by $g(\theta)$ and differentiate with $\theta$ instead of $\hat{y}$. In the last two lines, $\partial_\theta g(\theta)$ appears in front of both integrals.

$$
\begin{aligned}
0 &= \frac{\partial}{\partial\theta}\mathbb{E}[l_\alpha(Y, g(\theta))] \\
&= \frac{\partial g(\theta)}{\partial\theta}\left[(1-\alpha)\int_{-\infty}^{g(\theta)}\,\mathrm{d}\mu_Y(y) - \alpha\int_{g(\theta)}^\infty\,\mathrm{d}\mu_Y(y)\right] \\
&= \frac{\partial g(\theta)}{\partial\theta}\left[(1-\alpha)F_Y(g(\theta)) - \alpha(1 - F_Y(g(\theta)))\right]\ .
\end{aligned}
\tag{3}
$$

Calculating the Euclidean norm of this vector equation, we obtain

$$
\|\frac{\partial g(\theta)}{\partial\theta}\|\cdot|(1-\alpha)F_Y(g(\theta) - \alpha(1 - F_Y(g(\theta))| = 0
\tag{4}
$$

Since $\partial g(\theta)/\partial\theta \neq 0$, we can divide both sides by its norm, yielding $g(\theta) = F_Y^{-1}(\alpha)$. $\qquad\square$

**Remark 4.** *Since the only property of a minimum we have used when deriving Theorem 1 and Corollary 2 is that the first derivative is zero in the minimum, $\hat{y}$ (or $\theta$) could be also a local minimum, local maximum or even a saddle point.*

## 4 Connection to problem statement

The result of the previous section can be connected with the problem stated in Sec. 2 by considering the dependence of the predicted variable $Y$ on features $X$ (which include also the time variable). Assume that we have found such a model parameter set $\theta$ that the expected value of the loss function $l$ is minimized for every value of $X$,

$$
\frac{\partial\mathbb{E}[l(Y, g_\theta(X))|X]}{\partial\theta} = 0
$$

for any $X$. Then, the results of Section 3 can be easily modified to show that $g_\theta(X) = F_{Y|X}^{-1}(\alpha)$, by replacing $g(\theta)$ with $g_\theta(X)$, $\mu_Y$ with $\mu_{Y|X}$ (distribution of $Y$ conditional on $X$) and $F_Y$ with $F_{Y|X}$ (CDF of $Y$ conditional on $X$), and the assumption $\partial g(\theta)/\partial\theta \neq 0$ for all $\theta$ with the assumption $\partial g_\theta(X)/\partial\theta \neq 0$ for all $\theta$ and $X$.

Standard model training, however, does not minimize $\mathbb{E}[l(Y, g_\theta(X))|X]$, but $\mathbb{E}[l(Y, g_\theta(X))]$ (as mentioned above, averaging over all $X$ in a large training set), which is the quantity considered by its proofs of convergence Shalev-Shwartz & Ben-David (2014). The condition $\frac{\partial}{\partial\theta}\mathbb{E}[l(Y, \hat{y})] = 0$ leads then to

$$
\begin{aligned}
0 &= \frac{\partial}{\partial\theta}\mathbb{E}[l(Y, \hat{y})] \\
&= \int_{\mathcal{X}}\frac{\partial g_\theta(x)}{\partial\theta}\left[(1-\alpha)F_{Y|X=x}(g_\theta(x)) + \alpha(1 - F_{Y|X=x}(g_\theta(x)))\right]\,\mathrm{d}\mu_X(x),
\end{aligned}
\tag{5}
$$

where $\mu_X$ is the distribution of feature vectors $X$.

Consider now a model with a separate constant term, $g_\theta(x) = G(\xi)$, where $G(\xi)$ is strictly monotonic function of $\xi = \theta_0 + f_\theta(x)$. Most neural network prediction models are indeed of such a form. The component of Eq. (5) for $\theta_0$ has the form

$$
\begin{aligned}
0 &= \frac{\partial G}{\partial\xi}\int_{\mathcal{X}}\left[(1-\alpha)F_{Y|X=x}(g_\theta(x)) + \alpha(1 - F_{Y|X=x}(g_\theta(x)))\right]\,\mathrm{d}\mu_X(x) \\
&= \frac{\partial G}{\partial\xi}\left[(1-\alpha)\mathbb{E}[F_{Y|X}(g_\theta(X))] + \alpha(1 - \mathbb{E}[F_{Y|X}(g_\theta(X))])\right]\ .
\end{aligned}
\tag{6}
$$

Hence, we have $\mathbb{E}[F_{Y|X}(g_\theta(X))] = \alpha$ since $\partial G/\partial\xi \neq 0$.

We would like to make a stronger statement, that is, $F_{Y|X}(g_\theta(X)) = \alpha$ (note that, given the above, $\partial g_\theta(X)/\partial \theta \neq 0$ for all $\theta$ and $X$), but the integration over $X$ precludes it. To support this conjecture we indicate the following heuristic arguments:

1. Recall that stochastic gradient descent (SGD) uses mini-batches to compute the loss gradient. By training long enough with small enough mini-batches sampled *with replacement* (to generate a greater variance of mini-batches), we can hope to at least approximately ensure that $\mathbb{E}[l(Y, g_\theta(X)]$ is minimized "point-wise" w.r.t. to $X$. This is supported by the fact that

$$\mathbb{E}[l(Y, g_\theta(X))] = \int_{\mathcal{X}} \mathbb{E}[l(Y, g_\theta(X))|X = x]\rho_X(dx)$$

   On top of that, section 3 in Zhou et al. (2019) and Theorem 1 in Turinici (2021) suggest that SGD should still work for loss functions of the form (1) for reasonable network choices.

2. If the model $g_\theta$ is flexible enough that it is capable of globally minimising $\mathbb{E}[l(Y, g_\theta(X))|X]$ for every $X$ (for example, it is produced by a very large deep neural network), then this global pointwise minimum has to coincide with the global "average" minimum of $\mathbb{E}[l(Y, g_\theta(X)]$ and can be found by training the model long enough. Formally, if there exists such a $\hat\theta$ that $\hat\theta = \arg\min_\theta \mathbb{E}[l(Y, g_\theta(X))|X]$ for all $X$, then also $\hat\theta = \arg\min_\theta \mathbb{E}[l(Y, g_\theta(X)]$.

3. Recall that $\partial g_\theta(X)/\partial \theta$ is a vector, not a scalar. Therefore, Eq. (5) is actually a set of conditions of the form $\mathbb{E}_X[A_i(X)B(X)] = 0$, where $A_i(X) = \partial g_\theta(X)/\partial \theta_i$ and $B(X) = (1-\alpha)F_{Y|X}(g_\theta(X)) + \alpha(1 - F_{Y|X}(g_\theta(X))$. The larger the neural network, the more conditions we have on the pairs $A_iB$, but they can be all automatically satisfied if $B(X) \equiv 0$. Moreover, the value of $B$ depends only on the predictions $g_\theta(X)$ produced by the model and the distribution of $Y$ given $X$ (which is dictated by the data distribution $\mathcal{D}$). On the other hand, the derivatives $\partial g_\theta(X)/\partial \theta_i$ depend also on the internal architecture of the neural network (for example, the type of activation function used, number of layers, etc). If, what is likely, there exists multiple architectures equally capable of minimising the expected loss, Eq. (5) must be true for all corresponding functions $\partial g_\theta(X)/\partial \theta_i$ at the same time, which also suggests $B(X) = 0$.

4. Consider now the limit of a very large, extremely expressive neural network, which achieves a global minimum of $\mathbb{E}[l(Y, g_\theta(X)]$. We can describe it using the following parameterisation: let $\theta = [\theta_i]_{i=1}^N$ and $g_\theta(x) \approx \sum_{i=1}^N \theta_i k_N(x - x_i)$, where $x_i$ belongs to the support of the distribution of $X$, $\int k_N(x - x_i)\rho_X(dx) = 1$, and $k_N(d) > 0$ approaches the Dirac delta distribution in the limit $N \to \infty$. We have $\partial g_\theta(x)/\partial \theta_i = k_N(x - x_i)$, and, for each $i$,

$$0 = \int_{\mathcal{X}} \frac{\partial g_\theta(x)}{\partial \theta_i} \left[(1-\alpha)F_{Y|X=x}(g_\theta(x)) + \alpha(1 - F_{Y|X=x}(g_\theta(x))\right]\rho_X(dx)$$
$$\approx \int k_N(x - x_i)\left[(1-\alpha)F_{Y|X=x}(g_\theta(x))\right.$$
$$\left. + \alpha(1 - F_{Y|X=x}(g_\theta(x))\right]\rho_X(dx) .$$

   In the limit of $N \to \infty$, $k(d) \to \delta(d)$ and the above integral becomes

$$0 = (1-\alpha)F_{Y|X=x_i}(g_\theta(x_i)) + \alpha(1 - F_{Y|X=x_i}(g_\theta(x_i)) \quad \text{for all } i.$$

   At the same time, as $N \to \infty$, the feature vectors $x_i$ have to progressively fill in the entire support of the distribution of $X$, hence the above condition is asymptotically equivalent to

$$0 = (1-\alpha)F_{Y|X=x}(g_\theta(x)) + \alpha(1 - F_{Y|X=x}(g_\theta(x)) \quad \text{for all } x \in \mathcal{X},$$

   or simply $F_{Y|X}(g_\theta(X)) = \alpha$.

5. As the opposite case to the above, consider a naive model $g_\theta(X) = \theta_0 + \theta_1 \cdot X$ describing a linear process, $Y = y_0 + w \cdot X + Z$, where the error term $Z \perp X$ is i.i.d. and $y_0, w$ are unknown constants. From Eq. (6) we have $\mathbb{E}[F_{Y|X}(g_\theta(X))] = \alpha$. Writing Eq. (5) for $\theta_1$, we obtain

$$0 = \int x \left[(1 - \alpha)F_{Y|X=x}(\theta_0 + \theta_1 \cdot x) + \alpha(1 - F_{Y|X=x}(\theta_0 + \theta_1 \cdot x))\right] \rho_X(dx)$$

$$= \int x \left[(1 - \alpha)F_Z(\theta_0 - y_0 + (\theta_1 - w) \cdot x)\right.$$
$$\left. + \alpha(1 - F_Z(\theta_0 - y_0 + (\theta_1 - w) \cdot x))\right] \rho_X(dx)$$

It is easy to see that both conditions are satisfied by $\theta_1 = w$ and $\theta_0 = F_Z^{-1}(\alpha)$. Then, we have $F_{Y|X}(g_\theta(X)) = \alpha$ for all $X$. Hence, our method works perfectly in the linear case with i.i.d. error terms.

## 5 Experiments and results

In this section, we present the results of two case studies: the first is forecasting the daily demand for 15 different food products, and the second is forecasting revenue for 10 food stores that sell them. We utilize historical data to back-test our forecasts with PIs.

The core forecasting model is an adaptation of N-BEATS architecture Oreshkin et al. (2019), whose initial version was developed by Molander Molander (2021). For each study, we train the model simultaneously on all training data in the spirit of global learning approach. The product demand dataset consists in time series of daily product sales of varying length spanning up to 4 years, product name and category, weather information, a custom calendar indicating special days and periods, time of selling out, as well as geographic location and store type. The revenue dataset includes time-series of revenues for the stores, along with information on store type, location, weather, and the custom calendar. The features are represented as integers, one-hot-encoded vectors or (trainable or not) word embeddings. They have adjustable training weights and can be excluded by the hyperparameter tuning. Custom sample boosting techniques support better performance on the special calendar days. The model also attempts to detect anomalies in sale trends resulting from Covid-related lockdowns.

During the training, mini-batches are drawn with replacement, following point 1 of Sec. 4. The Tensorflow implementation of our model uses the Adam optimiser Kingma & Ba (2014) with early stopping and learning rate decay (starting from $10^{-3}$). The algorithm converges in less than 70 epochs.

We optimise a multitask loss function, which is the sum of three distinct loss functions defined by Eq. 1, $l(\alpha = 0.5) + l(\alpha = 0.5 - \beta/2) + l(\alpha = 0.5 + \beta/2)$, thus estimating the result and the PIs of a desired probability level $\beta$ at once. Training a single neural network to predict all three values simultaneously is not only more computationally efficient, but also helps capture their dependencies, thereby avoiding stochastic artefacts (such as crossing PIs, which we occasionally observed when estimating the three values in separate training runs).

The model utilizes a 14-day window of recent data, weather forecasts, and upcoming events to predict the next day's sales or revenues. We limit the forecast horizon to 1 day, as per the theoretical assumptions. Therefore, for the period from 2 April to 21 July, we generate 111 predictions per product, which we back-test against the historical data to evaluate our model's performance. In addition, we experiment with longer forecast horizons of one or two weeks (which utilise additional extensions to the model to preserve theoretical correctness and efficacy), demonstrating the stability of our method.

Figures 1 and 2 show the results of the two case studies: product demand and store revenue predictions with PIs, respectively. The median demand was calculated together with the 70% PI using the multiloss function described above; in addition, the 80% and 90% PIs are plotted. The interval width grows with increasing probability level, as expected. (The largest widening of the prediction interval should occur just before reaching 100%, assuming the errors follow either a Laplacian or Gaussian distribution.)

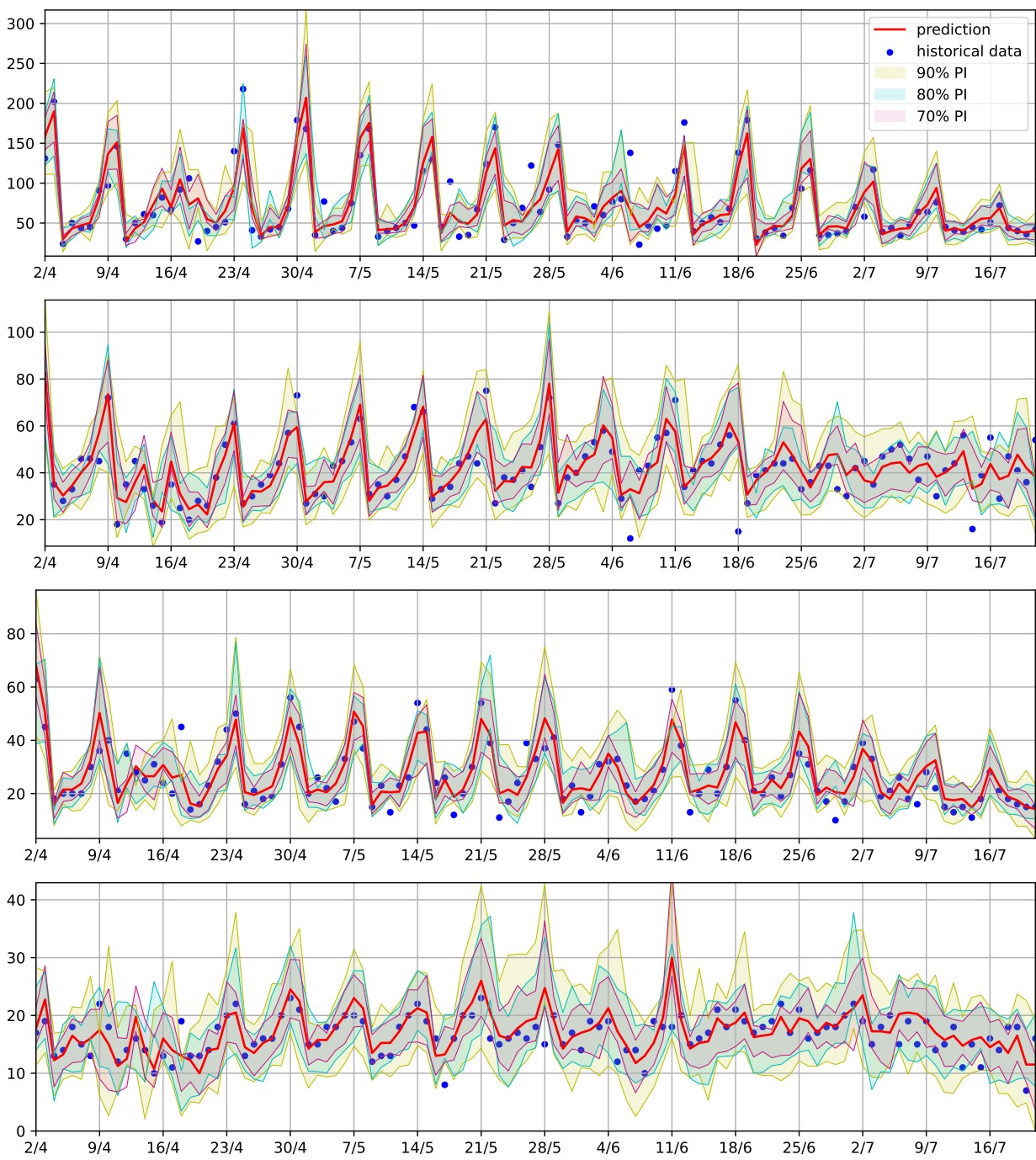

Figure 1: Product sale forecasts with 70%, 80% and 90% PIs for four products calculated using multi-loss function (median shown calculated with 70% PI.

While the widths of the prediction intervals generally remain consistent over time, there are some exceptions. A typical example is the spike of upper PI on 1 May for the first product: an unusually high sale on the previous week's same weekday, combined with a periodic (weekly) sales behaviour, caused the model to increase the uncertainty. As a result, the historic sales data points are indeed captured within the interval.

As the probability level increases, the width of the interval grows larger, based on our test results for 70%, 80% and 90% PIs. Assuming the errors follow either a Laplacian or Gaussian distribution, we expect the

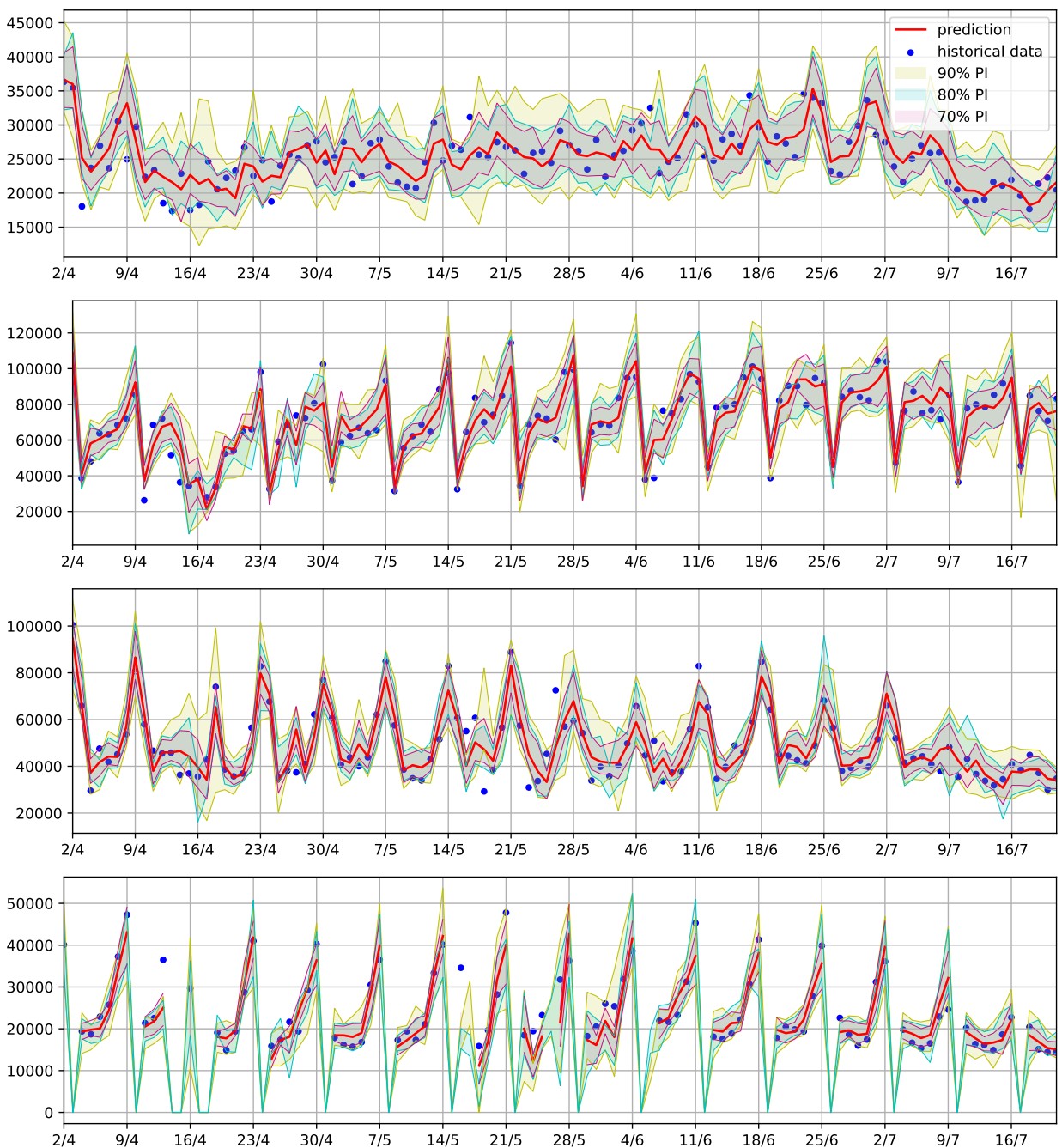

Figure 2: Revenue forecasts with 70%, 80% and 90% PIs for four stores using multi-loss function (median shown calculated with 70% PI.

largest widening of the prediction interval to occur just before reaching 100%, with a theoretical widening of the PI to infinity, as the probability approaches 100%.

The PI coverage probability for the three tested widths is shown in Table 1. The proportion of future realised values that fall within the PIs was counted for all daily forecasts of 15 product demand and 10 store revenues. Equal or slightly lower than the PI probability level, the results suggest that indeed our method accurately finds lower and upper bounds on the probability distribution of forecasted quantity. We note the coverage is

virtually perfect when we train the model of the same size to predict different percentiles separately, instead of using the multiloss function (which is more efficient for our purposes).

Table 1: The prediction interval coverage probability for tested PI widths of daily forecasts.

| PI width | PI coverage |
|----------|-------------|
| 0.9 | 0.89 |
| 0.8 | 0.8 |
| 0.75 | 0.74 |
| 0.7 | 0.67 |

We additionally experiment with longer forecast horizons required for the practical applications of our model. Figure 3 shows examples of forecasts with 7-day and 14-day horizons, i.e. performed every 7 or 14 days, respectively. Quite surprisingly, the success-rate on our set of tested products remains high, namely 76% of historical points lie within the analysed 7-day 75% PI and 74% of historical points lie within the 14-day 75% PI. The starting and final days of the forecasts are marked by the grid, where they overlap. As expected, at these points we can see that the PI widens towards the end of the predicted period and narrows erratically as the new prediction period begins.

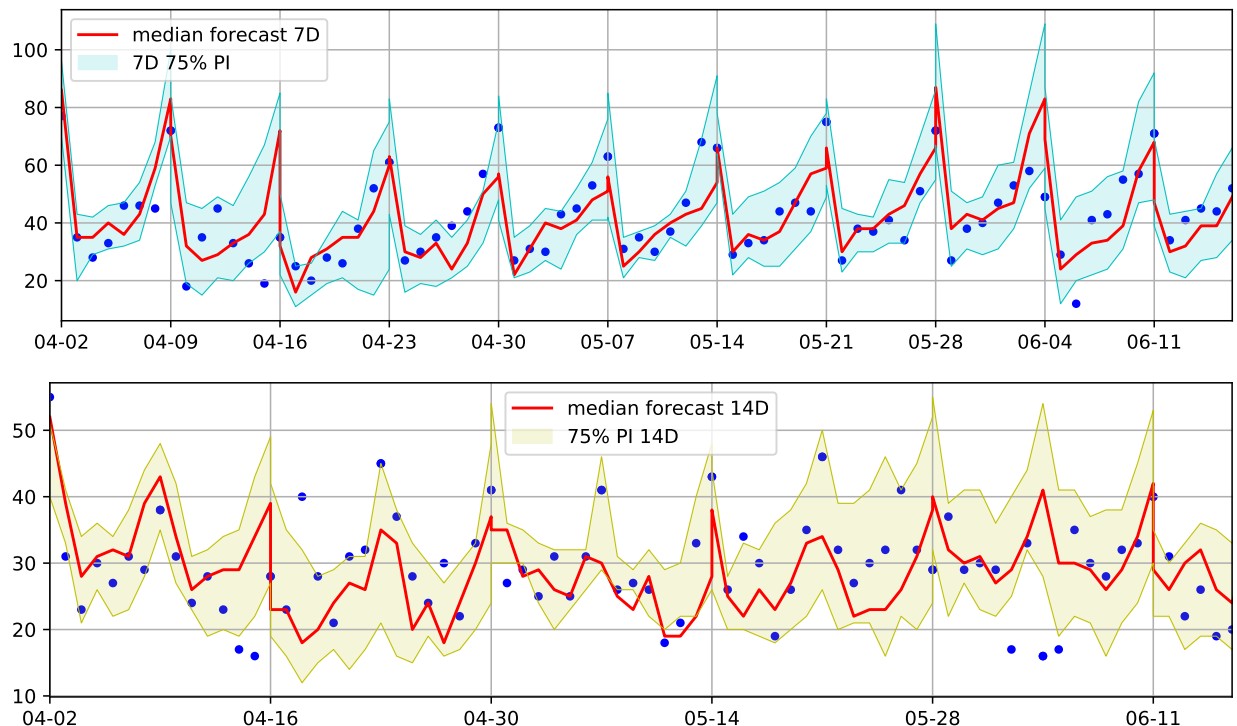

Figure 3: Sale forecasts with 7 and 14-day horizon trained by MAE with 70% PIs, performed every 7 or 14 days, respectively.

# 6 Final thoughts and future work

## 6.1 Final thoughts

The standard interpretation of regression models relies on maximum likelihood—their estimate is the best they can give based on the available data. Without additional *ad hoc* (as reflected in Remark 1) assumptions about the data distribution, it does not even define what statistics they actually estimate. Neither does it

endow us with an intuition about the quality of the estimate or the models' predictive power. In fact, it can be misleading, as increasing the likelihood does not necessarily mean increasing the model accuracy, vide overfitting.

Statistical learning theory provides a more general answer to the question "what does a nonlinear regression model predict?". Minimising the expected loss over a sufficiently large i.i.d. training set, we obtain with high probability a model producing, on previously unseen test data, predictions with the expected loss not much larger than the minimal expected loss possible for given model architecture Shalev-Shwartz & Ben-David (2014). The only condition is that the test data are drawn from the same distribution as the training set.

Yet, the actual task of statistical regression models is to provide the range of values that likely contains the true outcome, given the trained parameters and the data. We presented a simple method for training a neural network to predict lower or upper bounds of the PI based on a vector of features. It does it by minimising a weighted version of the MAE loss, the only assumption being that the model is expressive enough to obtain an (approximate) global minimum of the expected loss. We have shown through numerical experiments that this mathematical result holds true in real-world settings, which suggests that our method can be used to create PIs of neural networks in similar setups.

### 6.2 Future work

Although we have demonstrated that the proposed method works for linear models and expressive models, it may be less effective in specific cases. Identifying these weaknesses can help improve our understanding of the method's potential applications.

The relationship between the optimal model parameters $\theta$ and hyperparameters is generally complex and not straightforward. It is worthwhile to explore the relationship between the percentile $\alpha$ and the resulting model parameters $\theta$, particularly in terms of continuity and regularity.

Using different asymmetric loss functions may lead to other interesting minimizations. We conjecture that a similar result exists for a weighted MSE minimisation, where the predicted value is related to the amount of deviation from the prediction found by minimising MSE. This should be directly related to the chosen weight $\alpha$.

Finally, our method works well in practice even though its theoretical requirements are not guaranteed to be satisfied by the standard model training algorithm (see Sec. 4). This surprising finding warrants further investigation, which could lead to interesting insights into how neural networks learn during training.

## 7 Acknowledgements

We would like to thank Professor Erlend Aune for the suggestion to use a multitask loss, which helped us improve the computational efficiency.

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
