# OpenReview forum: "Prediction interval for neural network models using weighted asymmetric loss functions."
_TMLR — Rejected by TMLR_

### Review · Reviewer_1VE1 · 2023-07-28

**Summary Of Contributions:**

The paper studies the problem of learning machine learning models with reliable prediction intervals, i.e. intervals that contain the true label with high probability. The paper assumes an additive noise on the predicted label that follows the Laplace distribution. It then derives a simple loss function from these assumptions.

**Audience:**

No

**Broader Impact Concerns:**

While the paper briefly addresses the limitation of the proposed method in Section 6.2, it does not sufficiently address the broad impact of their solution, i.e. how their findings may influence other domains

**Claims And Evidence:**

No

**Requested Changes:**


The paper should take a detailed look at the uncertainty calibration literature, describe the similarities and differences of the proposed method to the existing methods, clarify the advantages of the differences of the proposed method, and make a numerical comparison to them in sufficiently advanced data sets and show either a qualitative (e.g. acquisition of a new source of knowledge) or a quantitative (e.g. higher prediction accuracy) improvement against them. Honestly I have major doubts that the proposed method in its current shape can pass this test. It is very likely that elaborating also on the novelty part, i.e. the whole story line, will be necessary. All this sounds to me like repeating the whole scientific development process of the work from scratch

**Strengths And Weaknesses:**

The paper studies the problem known well in the machine learning literature as “uncertainty calibration”. However, it totally ignores the related literature. A starting point could be:

C. Guo et al., On Calibration of Modern Neural Networks, ICML, 2017

For regression and time series prediction one may see:

V. Kuleshov et al., Accurate uncertainties for deep learning using calibrated regression, ICML, 2018

P. Cui et al, For regression and time series prediction, NeurIPS, 2020

For dynamical systems modeling, the following is a good source:

A. Look et al., A deterministic approximation to neural SDEs, PAMI, 2022

The experiment results reported in the paper are significantly behind the level of the state of the art. This fact will be obvious to the authors when they take a careful look at the experiments reported in the papers I listed above.

The proposed methodology is rather straightforward. Theorem 1 is a trivial fact. The choice of the Laplace distributed noise is unjustified and probably even unnecessary.

---

### Review · Reviewer_pT7W · 2023-07-29

**Summary Of Contributions:**

This paper shows that weighted assymetric losses are a simple approach to generate
prediction intervals for forecasting problems. The weighted assymetric loss is simple
to specify only requiring the percentile value $\alpha$.

The paper starts with a simple mean
estimation problem problem and shows that the optimal solution of the loss function converges to the true value of the percentile. This is extended to parametrised functions and even for
conditional forecasts. The assumptions and proofs are clearly described in the paper.

Experiments on benchmark datasets show that the coverage probabilities of the confidence
intervals match the intended coverage $\alpha$.


**Audience:**

Yes

**Claims And Evidence:**

Yes

**Requested Changes:**

Addition of baselines and comparison using the CRPS metrics are needed for a more comprehensive evaluation of this work.

**Strengths And Weaknesses:**

The main drawback of the paper is the proposed loss function is exactly the same as the already well known Pinball loss.
As such the novelty of this paper is unclear. Although the proofs for the case of parametrised and conditional distributions may be new, they follow from relatively simple derivations.

This loss is applied to a variety of publicly available forecasting problems and the experimental results look promising. Empirical estimates of the coverage probability match the intended percentile values. However, this metric does not talk about the quality of the predictions (see [1] for more context). In fact, the coverage probabilities can be exact even for a properly chosen constant value which does not change with the input features. In addition to the coverage probabilities, the CRPS metric would be useful in judging the quality of the predictions.

The paper does not include any strong baselines either for comparison.

[1] Probabilistic Forecasts, Calibration and Sharpness, Gneiting et. al.

---

### Review · Reviewer_kvFy · 2023-08-05

**Summary Of Contributions:**

This paper studies the problem for predicting confidence intervals around the model predictions. It proposes to minimize weighted mean-absolution error as the loss function (which is asymmetric around 0). It describes a simple and efficient method to estimate the upper and lower bounds of the predicted intervals (PIs). Finally, the paper shows some theory to justify why the proposed method works for predicting PIs of dependent variables.

**Audience:**

Yes

**Claims And Evidence:**

Yes

**Requested Changes:**



Questions for Authors:
------------
- How do you justify the choice of the multi-task loss used in the experiments section? l(alpha=0.5) + l(alpha=0.5-beta/2) + l(alpha=0.5+beta/2)
- Do you have any comparison with baselines such as LUBE?
- Have you tried other loss functions than MAE? Other regression loss function such as MSE or Huber loss, etc.?


Nit-Picks:
--------

- Title : Remove '.' at the end "Prediction interval for neural network models using weighted asymmetric loss functions."
- Put citations in paranthesis for reading/writing clarity. for example: Sec.1: ".. weather Liu et al. (2021), water level Zhang et al. (2015), price Duan et al. (2022), electricity grid load Rana et al. (2013), ecology Miok (2018), demographics Werpachowska (2018) or ..". Currently citations without paranthesis makes the text a bit harder for reading.
- Please do not include acknowledgements (Sec.7) in the manuscript submitted for review (you should add this section once the manuscript has been accepted). This affects double blind reviewing process.
- Sec.2.Problem-Statement: ".. where $\theta$ is are .. "

**Strengths And Weaknesses:**

Strengths:
-----------
- Simple asymmetric weightes loss helps in estimating the upper and lower confidence intervals of the model prediction

Weaknesses:
-----------
- Lack of comparison with other baselines that perform confidence interval predictions
- Method applied only on one dataset, need more thorough evaluation of the proposed scheme to state that it can predict the upper and lower confidence intervals efficiently in wide setup.

---

### Decision · Action_Editors · 2023-09-21

**Recommendation:** Reject

**Comment:**

Unfortunately the reviewers unanimously agreed to reject the paper.  While the methodology seems correct and the presented experiments look promising, the paper is incomplete.  I don't mean to discourage the authors, the presented idea seems promising.  However, publishing in the field requires the author to present their work in the context of the existing literature.  How can we progress as a field if every method is presented in isolation without consideration of what has come before?  There were also questions about the novelty of the method which should be addressed.  I encourage the authors to run experiments with appropriate comparisons as baselines, on competitive benchmarks and consider submitting to a future venue.  However, the amount of additional work is significant, so I wouldn't recommend a revise and resubmit at this time.

**Audience:**

Yes, lightweight prediction of model uncertainty is certainly of significant interest to the machine learning community.

**Claims And Evidence:**

Two of three of the reviewers found that the claims were not supported by evidence.  One reviewer thought the claims were supported.  An issue here is that the paper presents a new method in a highly studied area of the literature, but offers no comparisons to related methods.

**Resubmission Of Major Revision:**

The authors may consider submitting a major revision at a later time.